## Article

# Optimal Power Allocation in Optical GEO Satellite Downlinks Using Model-Free Deep Learning Algorithms

Theodore T. Kapsis, Nikolaos K. Lyras and Athanasios D. Panagopoulos *

School of Electrical and Computer Engineering, National Technical University of Athens, GR15780 Athens, Greece; teokapsis@mail.ntua.gr (T.T.K.); lyrasnikos@central.ntua.gr (N.K.L.)
* Correspondence: thpanag@ece.ntua.gr

**Abstract:** Geostationary (GEO) satellites are employed in optical frequencies for a variety of satellite services providing wide coverage and connectivity. Multi-beam GEO high-throughput satellites offer Gbps broadband rates and, jointly with low-Earth-orbit mega-constellations, are anticipated to enable a large-scale free-space optical (FSO) network. In this paper, a power allocation methodology based on deep reinforcement learning (DRL) is proposed for optical satellite systems disregarding any channel statistics knowledge requirements. An all-FSO, multi-aperture GEO-to-ground system is considered and an ergodic capacity optimization problem for the downlink is formulated with transmitted power constraints. A power allocation algorithm was developed, aided by a deep neural network (DNN) which is fed channel state information (CSI) observations and trained in a parameterized on-policy manner through a stochastic policy gradient approach. The proposed method does not require the channels' transition models or fading distributions. To validate and test the proposed allocation scheme, experimental measurements from the European Space Agency's ARTEMIS optical satellite campaign were utilized. It is demonstrated that the predicted average capacity greatly exceeds other baseline heuristic algorithms while strongly converging to the supervised, unparameterized approach. The predicted average channel powers differ only by 0.1 W from the reference ones, while the baselines differ significantly more, about 0.1–0.5 W.

**Keywords:** atmospheric turbulence; deep reinforcement learning; deep neural network; free-space optical; optical satellite; power allocation; policy gradient; scintillation

## 1. Introduction

The progressively demanding criteria of fifth-generation (5G) mobile communications sparked the deployment of high-throughput short-range links and mesh topologies to cater to the increased user capacity and to reduce energy consumption [1,2]. The sub-10 GHz radio frequency (RF) band is almost exhausted and strictly regulated. Consequently, scientific research is moving towards the incorporation of millimeter and nanometer wavelengths [1,2]. Optical wireless communications (OWC) operate similarly to fiber optics by modulating a coherent laser beam that propagates point to point and by line of sight [2–6]. Free-space optical (FSO) technology has excellent backhauling capabilities of ultra-fast transfer of traffic between antenna towers and small cells [2–6]. FSO also mitigates the last mile problem, i.e., congestion in the component linking the user to the Cloud/Internet [2–6].

Satellite communications (SatComs) cover a large portion of the Earth, including remote locations, and enable a vast variety of forecasting and broadcasting applications [6,7]. Yet, the cost of designing and manufacturing them is substantial, satellite debris causes "space pollution", and a satellite constellation requires numerous satellites in addition to very quick handovers to guarantee visibility and continuity [6,7]. Especially for high-speed SatComs, FSO systems have exhibited great potential due to their easy installment (<30 min), their operation with low initial expense and maintenance, no licensing requirements like RF systems, and their secure connections due to their large antenna gains,

allowing several FSO links to be deployed in parallel and in proximity [2–6]. Nevertheless, they demand advanced pointing, acquisition, and tracking components. The process of creating a private encryption key between two parties is known as quantum key distribution (QKD) [7]. Fundamentally, QKD is an optical technology that can supply encryption keys for any two locations connected by an optical link [7]. The application of QKD over optical fiber, on the other hand, is restricted by exponential fiber losses. In this setting, QKD over satellites is becoming more popular. The capacity to put up robust optical links and guarantee a minimal quantum bit error rate (QBER) by overcoming numerous transmission obstructions is critical to the success of QKD over satellites [7].

However, the FSO beam is susceptible to various problematic atmospheric phenomena, i.e., absorption, scattering, cirrus clouds, and turbulence [2–6]. Physical obstructions, geometric losses, and the blockage of the link caused by cloud occurrence are some of the difficulties [8,9]. For mitigating the cloud occlusion, site diversity can be used [8]. Received irradiance scintillation is created from rapid changes in wind speed, pressure, and temperature, which in turn induce changes in the refractive index [10–12]. The impact of scintillation depends primarily upon the time of day, the elevation angle of the link, and the altitude of the station [10–12]. In the daytime, at low elevation angles and low-altitude stations (denser atmosphere), turbulence is more extreme [10–12].

In [10–12], the power allocation (PA) problem is investigated for optical satellite downlinks under weak turbulence and solved using a Karush–Kuhn–Tucker (KKT), water-filling-inspired algorithm. However, the reported methodology depends on the knowledge of the system and channel model. In [13,14], a radio-on-FSO (RoFSO) wavelength multiplexing scenario is considered where the channel model is assumed unknown. The developed model-free primal–dual deep learning algorithm (PDDL) strongly converges to the precise solution derived from the model-based algorithm. In both studies, however, synthetic data were employed while the optical satellite-to-ground scenario was not explored. Likewise, in [15] a MIMO FSO system is studied and the PA problem is solved via reinforcement learning (RL) in a deep deterministic policy gradient (DDPG) approach. More power allocation problems regarding FSO, RF, and terrestrial and non-terrestrial networks exploiting ML and deep RL techniques can be found in [16–22].

In this paper, several channel-model-free methodologies and heuristics are explored for optimal PA, and then compared to the exact, model-based solution for an all-optical, multi-aperture satellite downlink between a geostationary (GEO) satellite and an optical ground station (OGS). Deep learning (DL) constitutes a powerful tool to handle data from complex and fading communication channels [23–25]. Therefore, the PA problem is formulated as a constrained learning optimization problem with peak and total expected power inequality constraints. Specifically, a deep reinforcement learning (DRL) module is proposed that assists the agents in producing actions via a stochastic policy gradient technique, given channel state information (CSI) observations from the environment. The policy is structured as a deep neural network (DNN) and trained according to the REINFORCE (REward Increment = Nonnegative Factor times Offset Reinforcement times Characteristic Eligibility) algorithm [26], but modified to include the power constraints, and the multi-agent optical environment where the agents act, independently accessing only their local observations but collaboratively trying to maximize the global reward. The proposed PA strategy is deemed appropriate for difficult optical conditions since it learns explicitly from observation. Whilst the optimization problem has been approached in the literature, it has never been presented in a MIMO optical satellite scenario along with available experimental data.

Summarizing the main contributions of this paper:

- We propose a DRL-aided algorithm to optimally allocate power in an optical GEO-to-ground multi-aperture system. The proposed method accurately adjusts the expected power for each optical channel, without any knowledge of path losses and scintillation conditions. Only CSI samples are utilized. The total expected power and peak power are constrained. Although in an LEO/MEO scenario deep learning would be even more beneficial, the GEO optical CSI is still unstable due to variations in the refractive

index structure parameter along the slant path, thin clouds that may attenuate or block the laser beam, and pointing and tracking errors. Thus, it is challenging to have a more accurate system model and fading distribution knowledge in the short and long term.

- Instead of simulated data, experimental irradiance time series from the European Space Agency's (ESA's) ARTEMIS optical satellite sessions were employed to evaluate the performance of the proposed methodology [27,28].
- The achieved ergodic system capacity from the application of the proposed algorithm greatly exceeds the performance of the model-free Equal Power, Random Power, and Deep Q-Learning Network (DQN) schemes [20–22] and approaches the model-based, unparameterized solution with very good agreement.
- An investigation of the impact of the number of hidden layers and neurons, policy distribution, and hyperparameter selection and overfitting effects was carried out.
- The proposed solution differs from other standard learning formulas because it applies to a multi-agent optical satellite PA problem based on the parameter sharing approach, allowing centralized learning under a single policy for faster convergence. The learning model is scalable as it has been tested in scenarios with a large number of optical satellite downlinks and a great amount of data and retained its performance. It is especially more scalable than the DQN algorithm because the Q-table is not scalable when there are large, high-dimensional, and continuous state–action pairs [20–22].

In Section 2, the optical carrier is described, the system model is presented, and the PA problem is formalized as a learning program. In Section 3, the proposed DRL-aided methodology is reported along with other heuristic, model-free strategies. In Section 4, the ARTEMIS mission is briefly discussed, and experimental measurements are employed to evaluate and compare the considered PA methodologies. Performance results are drawn and commented on. Section 5 concludes this work.

## 2. Channel Model and Power Allocation Problem

The turbulent cells of the atmosphere, known as eddies, work as a prism that will enhance or degrade a propagating optical signal [3]. If an eddy's diameter is almost the size of an incoming beam, it will result in received irradiance $I_r$ fluctuations called scintillation, which is the primary factor of deterioration in the FSO downlink [3]. The scintillation index (SI) $\sigma_I^2$ constitutes the normalized ratio of the standard deviation of received irradiance fluctuations to the mean received irradiance [3,29]:

$$\sigma_I^2 = \frac{\langle I_r^2 \rangle - \langle I_r \rangle^2}{\langle I_r \rangle^2} \tag{1}$$

where $\langle I_r \rangle$ represents the mean received optical irradiance in W/m². In satellite downlinks with $\sigma_I^2 < 1$ and elevation angle $> 20°$, the atmospheric turbulence is considered weak [10–12]. In weak turbulence conditions, the optical channel follows the Lognormal ($\mu$, $\sigma_I^2$) with Probability Density Function (PDF) [5]:

$$f_{I_r}(I_r) = \frac{1}{I_r \sqrt{2\pi}\sigma_I} \exp\left\{-\frac{\left[\ln(I_r/\langle I_r \rangle) + 0.5\sigma_I^2\right]^2}{2\sigma_I^2}\right\} \tag{2}$$

where $\mu = -0.5\sigma_I^2$.

The optical satellite downlink system under investigation is represented by the real-valued channel matrix $H \in \mathbb{R}^{N \times M}$ where $N$ is the number of ground receivers (Rx), and $M$ is the number of on-board transmitters (Tx). It is assumed that the covariance matrix $\Sigma = E\left[(H - E[H])(H - E[H])^T\right]$ has zero non-diagonal elements, i.e., linearly uncorrelated channels. In practice, this is achieved by having a distance between the Rx elements larger than Fried's parameter [1,29]. The CSI of the channels is denoted as $(h_{ij})_{1 \le i \le N, \, 1 \le j \le M}$

and the allocated power as $P_{ij}(h_{ij})$. The channel capacity is $C_{ij}(P_{ij}(h_{ij}), h_{ij})$. The average (ergodic) channel capacity is $\mathrm{E}[C_{ij}(P_{ij}(h_{ij}), h_{ij})]$.

For independent channels, and a proportionate number of Rx and Tx: $N = M = L$ and $i = j = l$, the total average capacity maximization problem is mathematically formalized as follows:

$$(\mathrm{OP}) \max_{\{P_l(h_l):\forall l\}} \sum_{l=1}^{L} \mathrm{E}[C_l(P_l(h_l), h_l)] \tag{3}$$

$$\text{s.t.} \quad 0 \le P_l(h_l) \le P_s, \quad \forall l$$
$$\sum_{i=1}^{L} \mathrm{E}[P_l(h_l)] \le P_{av} \tag{4}$$

where $P_s$ is the maximum safety power allocated to a single channel, and $P_{av}$ is the total available power constraint.

## 3. Proposed Methodology

### 3.1. The Exact Solution and the Supervised Approach

The Lagrangian function to maximize is [30]

$$L(P_l(h_l), v) = \sum_{\substack{l=1 \\ 0 \le P_l \le P_s}}^{L} \mathrm{E}[C_l(P_l(h_l), h_l)] - v\left(\sum_{l=1}^{L} \mathrm{E}[P_l(h_l)] - P_{av}\right) \tag{5}$$

where $v$ is the dual variable.

If the statistical models $f_{h_l}(h_l)$ are available, the average channel capacity is then [11,12]:

$$\mathrm{E}[C_l(P_l(h_l), h_l)] = \int_0^\infty \log_2\left(1 + \frac{P_l(h_l)|h_l|^2}{N_0}\right) f_{h_l}(h_l)\mathrm{d}h_l \tag{6}$$

and the average transmitted power is then [11,12]:

$$\mathrm{E}[P_l(h_l)] = \int_0^\infty P_l(h_l) f_{h_l}(h_l)\mathrm{d}h_l \tag{7}$$

The PA problem (3) and (4) is then convex [30]. The instantaneous optimal PA for the *i*th channel is given in [11,12]:

$$P_l^*(h_l) = \min\left\{P_s, \max\left(\frac{1}{v^*} - \frac{N_0}{h_l}, 0\right)\right\} \tag{8}$$

where $v^*$ is the optimal dual multiplier, and $N_0$ is the optical noise variance.

The $v^*$ is evaluated numerically from (4), (8) as follows:

$$\sum_{l=1}^{L} \int_0^\infty P_l^*(h_l) f_{h_l}(h_l)\mathrm{d}h_l = P_{av} \tag{9}$$

or via an iterative algorithm, e.g., sub-gradient method with decreasing step size $\eta$ as below [30]:

$$v^{k+1} = \left[v^k - \eta^k\left(P_{av} - \mathrm{E}\left[\sum_{l=1}^{L} P_l(h_l)\right]\right)\right]_+ \tag{10}$$
$$\eta^{k+1} = \eta^k/(k+1)$$

Note that the analytical expression (8) is valid as long as the $N_0$ is constant and not a function of $P_l(h_l)$. Also, (8) requires precise $h_l$, and (9) demands precise channel PDF knowledge to calculate $v^*$. In practice, it is very hard to assume accurate model information, CSI values, and constant optical noise.

Supervised learning is a process of providing labeled input datasets to the learning model to make predictions (regression) or decisions (classification) based on the labeled samples. In a regression problem, the output node is continuous, ranging theoretically from negative infinity to positive infinity. The mean square error (MSE) is the most common loss function to employ. From (8) and (10), the exact solution is provided (no need for a brute-force or exhaustive search), which can be utilized as ground truth to train a DNN to predict the optimal power allocation. Non-linear regression fitting or a multi-layer perceptron model may be implemented. However, this supervised approach is model-based.

*3.2. The Proposed DRL-Aided Algorithm*

DNNs consist of several layers of interconnected neurons and enable the learning of complex non-linear functions. RL teaches an agent how to execute actions upon entering a new state to maximize the cumulative future returns [23–25]. A DRL methodology refines its policy by directly calculating the policy gradient to maximize the cumulative rewards. The referred policy is a DNN that uses a continuous state as input and produces a probability distribution as output. A continuous action is then sampled. The core idea is to parameterize the policy using a DNN's weights parameter $\theta \in \mathbb{R}^w$ where $w = \sum_{i=1}^{Q} w_i w_{i+1}$ assuming a DNN with $Q$ layers, and each has a corresponding dimension $w_i$ [23–25].

Then, we denote the parameterized policy as the probability distribution of selecting power $P_{t,l}$ for the $l$th channel in time-step $t$, and from the state $(h_{t,l}, \theta)$ [13,14]:

$$\pi(h, \theta) = \Pr[P_{t,l} | h_{t,l}, \theta] \tag{11}$$

where $h_{t,l}$ represents recorded CSI data of $l^{\text{th}}$ channel, and $P_{t,l} \in [0, P_s]$ is the allocated power as derived from $\pi(h, \theta)$.

The agent's reward-return $r_{t+1}$ at the next time-step $t + 1$ is the following expression [13,14]:

$$r_{t+1} \underset{0 \le \pi(h_{t,l}, \theta) \le P_s}{= L(\theta, v)} = \sum_{l=1}^{L} C_{t,l}(\pi(h_{t,l}, \theta), h_{t,l}) - v\left(\sum_{l=1}^{L} \pi(h_{t,l}, \theta) - P_{av}\right) \tag{12}$$

Then, the discounted cumulative returns $G_t$ are defined as

$$G_t = r_{t+1} + \gamma r_{t+2} + \gamma^2 r_{t+3} + \ldots = \sum_{t'=t+1}^{T} \gamma^{t'-t-1} r_{t'} \tag{13}$$

where $T$ is the duration of an episode, $t = 0 \ldots T - 1$, and $\gamma \in [0, 1)$ is the discount factor [20–22]. The objective function is the long-term discounted rewards [13,14]:

$$J(\theta) = \mathrm{E}[G_t | \pi(h_t, \theta)] \tag{14}$$

In general, the gradient for a whole episode with duration $T$ is

$$\nabla_\theta J(\theta) = \int_T \nabla_\theta \pi_\theta(T) G(T) dT \tag{15}$$

If we apply the log-derivative trick $\nabla_\theta \pi_\theta = \pi_\theta \nabla_\theta \log \pi_\theta$ [13,14], then:

$$\nabla_\theta J(\theta) = \mathrm{E}[\nabla_\theta \log \pi_\theta(T) G(T)] \tag{16}$$

Then, we obtain a Monte-Carlo approximation of $\mathrm{E}[\,.\,]$ in (16) for all the $t = 0, 1, \ldots, T - 1$ time-steps and $l = 0, 1, \ldots, L - 1$ channels [31]:

$$\nabla_\theta J(\theta) = \frac{1}{T} \sum_{t=0}^{T-1} \sum_{l=0}^{L-1} [\nabla_\theta \log \pi(h_{t,l}, \theta) G_t] \tag{17}$$

To summarize, the proposed model-free, DRL methodology requires only $h_l$ samples that can be observed, $C_l$ values that can be experimentally measured, and a differentiable policy to be optimized. The proposed methodology is summarized in Algorithm 1.

---

**Algorithm 1.** DRL-aided Proposed Power Allocation Methodology

---

1: Initialize number of channels $L$, number of episodes $Ep$, episode duration $T$, learning rate $a$, discount factor $\gamma$, and weights $\theta$ of the policy network;
2:     **for** $i \leftarrow 0$ **to** $Ep - 1$ **do**
3:         Set $\nabla_\theta J(\theta) = 0$;
4:         **for** $t \leftarrow 0$ **to** $T - 1$ **do**
5:             **for** $l \leftarrow 0$ **to** $L - 1$ **do**
6:                 Fetch recorded CSI data $\{h_{t,l}\}$ for each $t$th time-step and $l$th optical channel;
7:                 Sample the allocated power $\{\pi(h_{t,l}, \theta)\}$ using the current policy distribution;
8:                 Calculate the returns $\{r_{t+1,l}\}$ from (12);
9:                 Calculate $G_{t,l} = \sum_{t'=t+1}^{T} \gamma^{t'-t-1} r_{t',l}$ from (13);
10:                $\nabla_\theta J(\theta) \leftarrow \nabla_\theta J(\theta) + \nabla_\theta \log \pi(h_{t,l}, \theta) G_{t,l}$ from (17);
11:             **end**
12:         **end**
13:         $\theta \leftarrow \theta + \alpha \nabla_\theta J(\theta)/T$ gradient ascent the policy parameter (DNN's weights);
14:         $v \leftarrow \left[ v - \alpha \sum_{t=0}^{T-1} \left\{ \sum_{l=0}^{L-1} \pi(h_{t,l}, \theta) - P_{av} \right\}/T \right]_+$   update dual variable (total power constraint);
15:     **end**

---

## 4. Simulation Results

In this section, the DRL-aided proposed PA methodology is applied to the experimental ARTEMIS time series data along with other heuristic, model-free strategies, i.e., DQN, Equal Power, and Random Power algorithms and the model-based supervised method.

The ESA's data relay satellite mission ARTEMIS (Advanced Relay and Technology Mission Satellite) took place in 2003 when the spacecraft attained GEO orbit, and involved a variety of optical communication telescopes for satellite-to-ground and intersatellite bi-directional links [27,28]. The experimental data to be utilized are recorded time series from the downlink transactions from the laser terminal (OPALE) onboard ARTEMIS to the reflector telescope (LUCE) installed on the Teide Observatory in Tenerife, 2400 m above sea level [27,28]. The spacecraft's location and communication features as well as LUCE's and OGS's essential characteristics are given in Table 1.

**Table 1.** ARTEMIS, OPALE, LUCE, and OGS Characteristics.

| Name | Characteristic | Value |
|---|---|---|
| ARTEMIS | Longitude | 21.5° East |
| OPALE | Latitude | 0.0° ± 2.81° North |
| | Altitude | 35,787 km |
| | Elevation Angle | 37° |
| | Coverage | Europe, Africa, and the Middle East |
| | Wavelength | 819 nm |
| | Beam Diameter ($1/e^2$) | 125 mm |
| | Transmitted Power | 10 mW |
| | Modulation Scheme | Intensity Modulation—Direct Detection |
| | Data Rate | 2 Mbps (downlink), 50 Mbps (uplink) |
| LUCE | Aperture Diameter | 26 cm |
| OGS | Altitude | 2.4 km |

In Figure 1, the normalized PDFs of five experimental time series are illustrated vs. the normalized PDFs of synthesized data that are lognormally distributed [27,28]. It is obvious that the PDFs derived from the retrieved ARTEMIS data accurately fit the lognormal PDFs,

implying weak turbulence conditions, and that the channel model in (2) can be applied to find the exact solution which will be used as labeled data for the supervised approach, elaborated in Section 3.1.

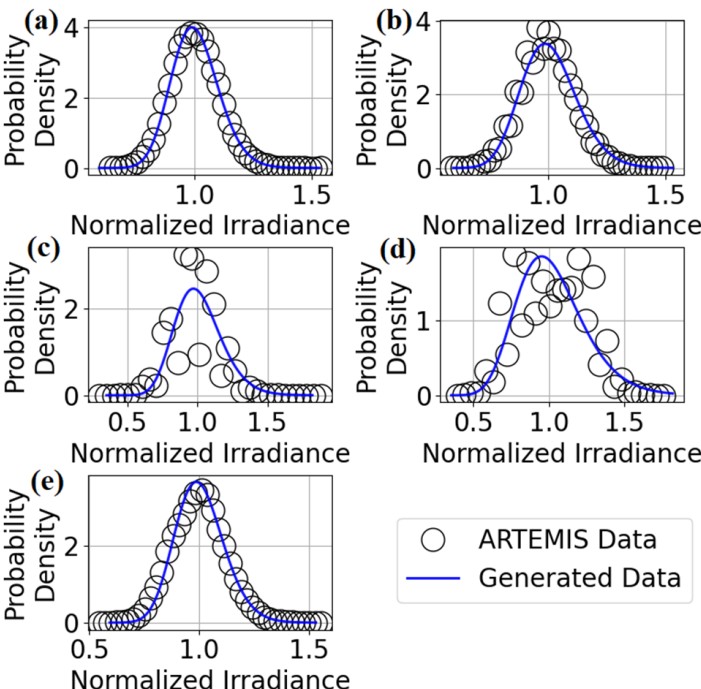

**Figure 1.** Normalized PDFs of experimental and synthesized data. (**a**) 10 September 2003 20:10–20:30. (**b**) 10 September 2003 00:30–00:50. (**c**) 12 September 2003 00:30–00:50. (**d**) 13 September 2003 23:30–23:50. (**e**) 16 September 2003 20:10–20:30.

The channel samples are then employed to form a $5 \times 5$ diagonal Multi-Input Multi-Output (MIMO) optical satellite-to-ground system. In general, the channels are assumed cloud-free but suffer from propagation losses and turbulence-induced scintillation.

In Table 2, the experimental optical channels' mean powers and SIs are included. The lower the $\langle P_r \rangle$ and the higher the SI, the worse the slant path conditions. The channels' correlation is negligible; hence, they are safely considered independent and will be denoted as ch.0, ch.1, ch.2, ch.3, and ch.4.

**Table 2.** Optical Channels' Parameters.

| Channel | Experiment | Parameters | Values |
|---------|------------|------------|--------|
| 0 | 10/09/2003 20:10 | $\langle P_r \rangle$, $\sigma_I^2$ | −13.34 dBm, 0.0101 |
| 1 | 10/09/2003 00:30 | $\langle P_r \rangle$, $\sigma_I^2$ | −15.79 dBm, 0.0142 |
| 2 | 12/09/2003 00:30 | $\langle P_r \rangle$, $\sigma_I^2$ | −19.78 dBm, 0.0274 |
| 3 | 13/09/2003 23:30 | $\langle P_r \rangle$, $\sigma_I^2$ | −20.94 dBm, 0.0485 |
| 4 | 16/09/2003 20:10 | $\langle P_r \rangle$, $\sigma_I^2$ | −13.85 dBm, 0.0121 |

To develop the proposed DRL-aided power allocation algorithm, the policy's DNN had to be accurately specified. In particular, the neural network's layers, nodes, and probability distribution must be chosen in a way to avoid overfitting, overhead, and high inference time. In Figure 2, the results of an investigation of the optimal number of hidden layers and neurons are shown. From Figure 2a, it is seen that one or two hidden layers are sufficient as they achieve the maximum average capacity and loss function. From Figure 2b, it is concluded that 256 neurons yield the best performance.

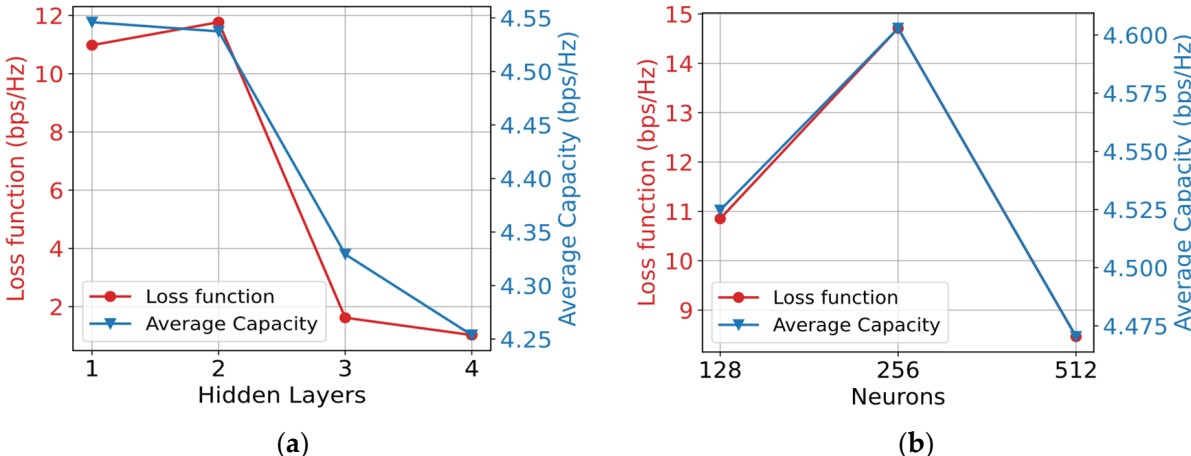

**Figure 2.** The loss function and predicted average capacity in terms of the (**a**) number of hidden layers (200 nodes/layer) and (**b**) number of neurons (2 hidden layers). $P_s = 1.0$ W, $P_{av} = 3.0$ W.

Finally, the stochastic policy that will determine the agents' actions was investigated. In Figure 3, three different probability distributions, i.e., truncated Normal $(\mu, \sigma, 0, P_s)$, truncated Weibull $(k, \lambda, 0, P_s)$, and truncated Exponential $(\lambda, 0, P_s)$ were tested and evaluated. Normal and Weibull are two-parameter distributions while Exponential is one-parameter. It is observed that the two-parameter distributions are the best choice and that truncated Normal performs slightly better.

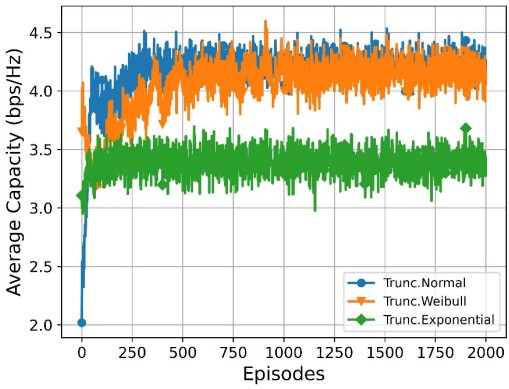

**Figure 3.** Performance results for five optical satellite downlinks using three different policy distributions: truncated Normal $(\mu, \sigma, 0, P_s)$, truncated Weibull $(k, \lambda, 0, P_s)$, and truncated Exponential $(\lambda, 0, P_s)$. $P_s = 1.0$ W, $P_{av} = 2.5$ W.

The finalized policy structure consists of a four-layer DNN, as reported in Table 3. In addition, the truncated Normal (TN) distribution is selected as a stochastic power policy for the five optical channels.

**Table 3.** Policy DNN Architecture.

| Channel | # Nodes | Activation |
|---------|---------|------------|
| Input | 5 | Linear |
| Hidden #1 | 200 | ReLu |
| Hidden #2 | 100 | ReLu |
| Output | 10 | Softplus |

The output nodes are used as parameters of the five TN distributions—means and standard deviations. The optimization algorithm involved in the training is RMSProp with step learning rate scheduling. Specifically, 5000 episodes with a duration of $T = 100$ time-steps

are considered to have frequent weight updates. The initial learning rate is set to 0.001 and the discount factor to 0.5 because fading negatively affects the impact of an agent's actions on his future expected returns [20]. Also, given the total power constraint, the agents' behaviors affect others' rewards due to the unpredictability of their neighbors' actions. Higher $\gamma$ is also undesirable because it decelerates the response to channel fluctuations [20]. The proposed algorithm was implemented in PyTorch.

In Figure 4, the predicted average capacity (a) and the constraint function (b) $\sum_{l=0}^{L-1} \mathrm{E}[P_l(h_l)] - P_{av}$ are depicted for $P_{av} = 2.0$ W. In Figure 5, the average channel powers are illustrated for the proposed, the supervised, the DQN, the Equal Power, and the Random Power algorithms for $P_{av} = 2.0$ W. Likewise, in Figure 6, the predicted average capacity (a) and the constraint function (b) are depicted for $P_{av} = 3.0$ W, and in Figure 7, the average channel powers are illustrated for $P_{av} = 3.0$ W. Finally, in Figure 8, the predicted average capacity (a) and the constraint function (b) are depicted for $P_{av} = 4.0$ W, and in Figure 9, the average channel powers are illustrated for $P_{av} = 4.0$ W. The DQN is a model-free, off-policy method that tries to predict the Q-values (expected future rewards) for every state–action pair using the $\epsilon$-greedy policy that alternates between exploration and exploitation [20–22]. DQN employs discretized actions: $\left\{0, \overline{P}/A - 1, 2\overline{P}/A - 1, \ldots, \overline{P}\right\}$ where A is the number of actions [20–22]. Here, A = 21 was chosen.

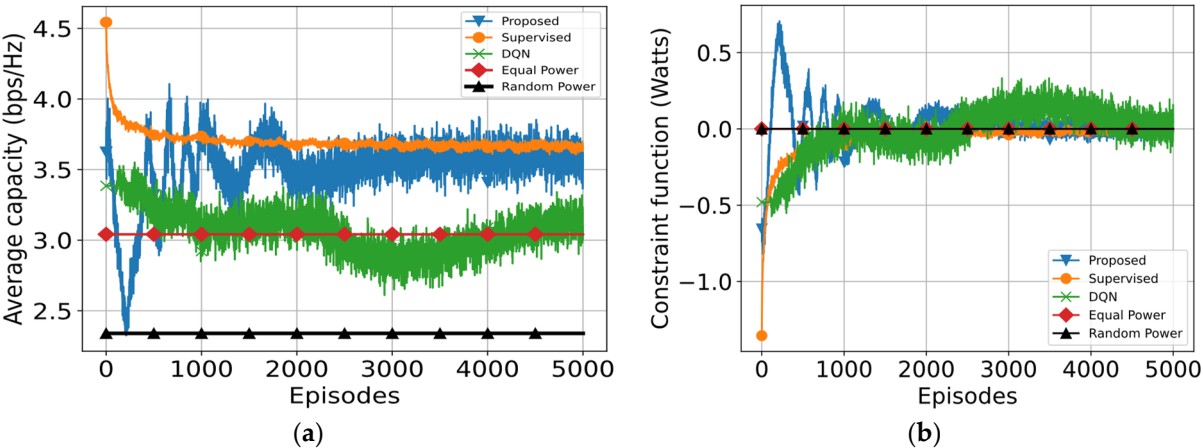

(a)　　　　　　　　　　　　　　　　　　(b)

**Figure 4.** Learning episodes of five different PA algorithms for five optical channels: (**a**) the average system capacity and (**b**) the constraint function. $P_s = 1.0$ W, $P_{av} = 2.0$ W.

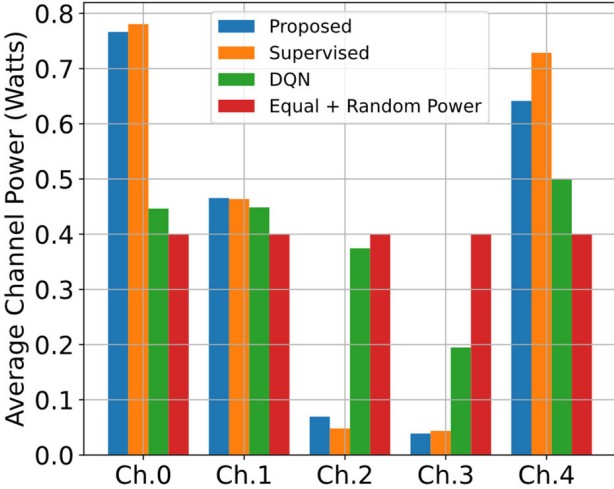

**Figure 5.** The average channel powers of five different PA algorithms for five optical channels during 5000 episodes. $P_s = 1.0$ W, $P_{av} = 2.0$ W.

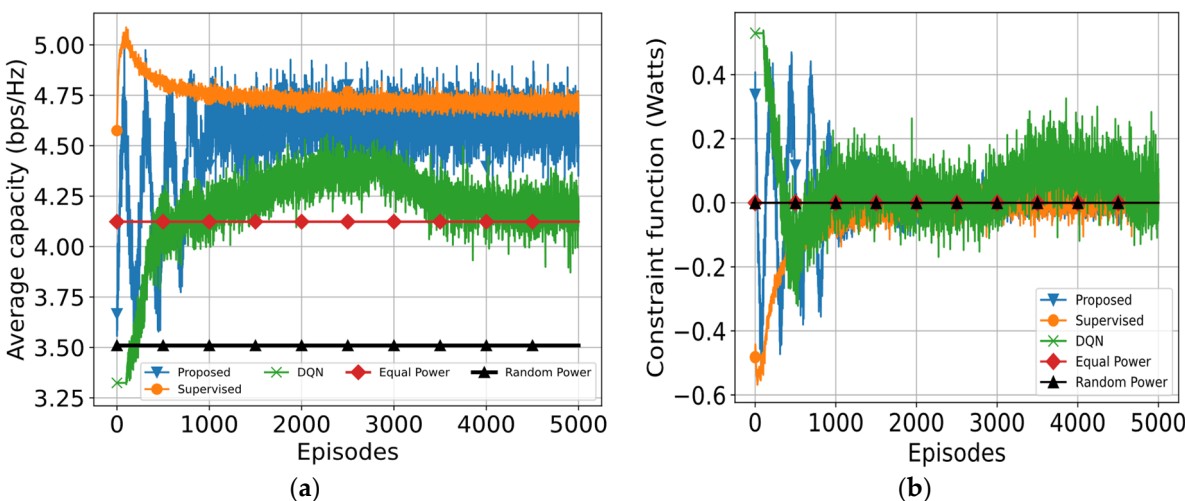

**Figure 6.** Learning episodes of five different PA algorithms for five optical channels: (**a**) the average system capacity and (**b**) the constraint function. $P_s = 1.0$ W, $P_{av} = 3.0$ W.

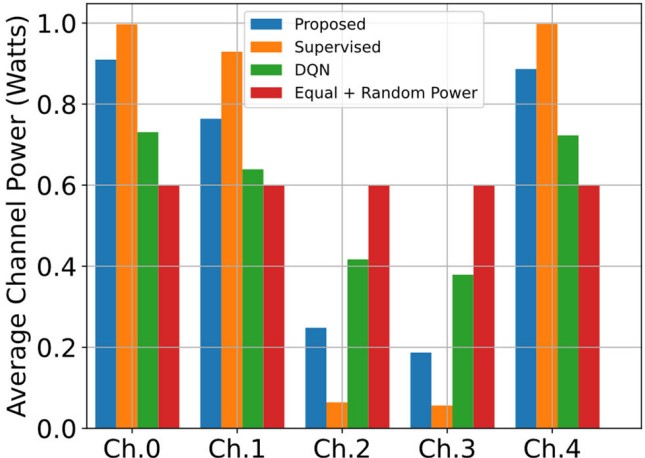

**Figure 7.** The average channel powers of five different PA algorithms for five optical channels during 5000 episodes. $P_s = 1.0$ W, $P_{av} = 3.0$ W.

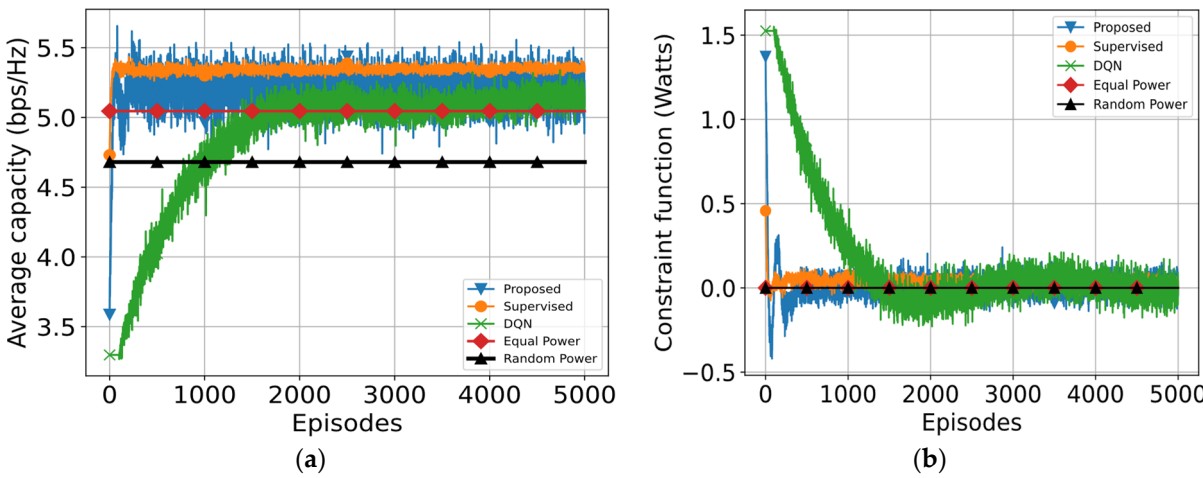

**Figure 8.** Learning episodes of five different PA algorithms for five optical channels: (**a**) the average system capacity and (**b**) the constraint function. $P_s = 1.0$ W, $P_{av} = 4.0$ W.

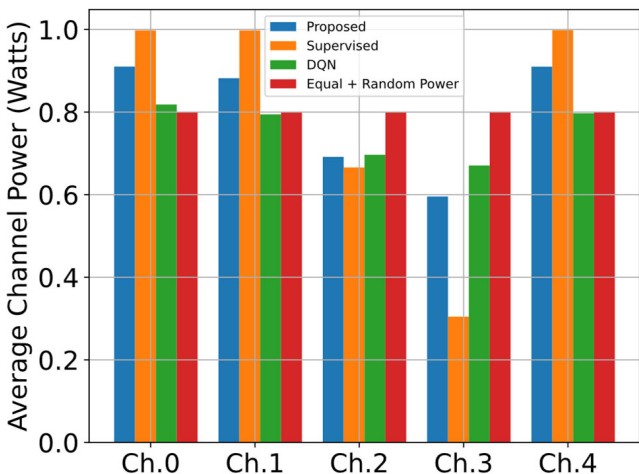

**Figure 9.** The average channel powers of five different PA algorithms for five optical channels during 5000 episodes. $P_s = 1.0$ W, $P_{av} = 4.0$ W.

The supervised algorithm was trained using the exact solution taken from the water-filling algorithm as if the channel statistics had been known to the transmitter. That scenario is expressed in Section 3.1 and incorporates (2), (8), and (10).

In Figures 4, 6 and 8, we observe that the proposed model-free DRL-aided solution outperforms the three baseline algorithms, showing a predicted average capacity advantage (18% better than Equal Power, 16% better than DQN, and 53% better than Random Power for $P_{av} = 2.0$ W). Additionally, it approximates the supervised method with excellent accuracy (2% worse performance for $P_{av} = 2.0$ W). The episode duration of $T = 100$ steps causes some waveform fluctuation, which can be resolved by using more time-steps at the cost of inference time. In Figures 5, 7 and 9, the average channel powers of the proposed algorithm match very well with the supervised ones, i.e., more power is assigned to the channels with better conditions, only differing by 0.1–0.2 W. The allocated channel powers of the other schemes differ significantly more, about 0.1–0.5 W. The inference time of an episode is 0.7–0.8 s including recorded data sampling, cumulative reward calculation, gradient computation, and back-propagation.

In Figure 10, the means and standard deviations of the TN distributions for ch.0 and ch.3 are depicted. Gradually, the values are stabilized according to their conditions in Table 2. The mean value of ch.0 (~1.2) is greater than that of ch.3 (~0.85), and the standard deviation of ch.0 (~0.2) is smaller than that of ch.3 (~0.5). Thus, ch.0 has a much higher probability of being allocated with $P_s = 1.0$ W than the weaker ch.3.

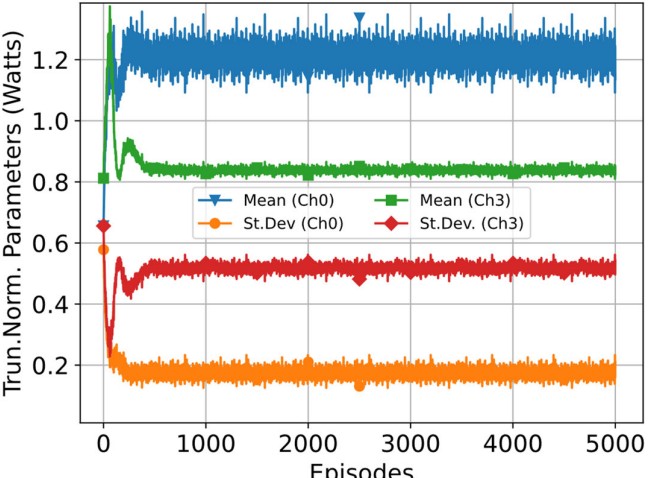

**Figure 10.** The mean and standard deviation of the TN distributions for two allocated channel powers corresponding to ch.0 and ch.3.

Finally, in Figure 11 the loss function is plotted for a training (two-thirds of original dataset) and a validation set (one-third of original dataset), and for two fixed hyperparameter $\lambda$ values. The $\lambda$ can be selected to adjust the trade-off between the power constraint violation and the loss function. The training loss is on the same level as the validation loss; therefore, no overfitting is observed.

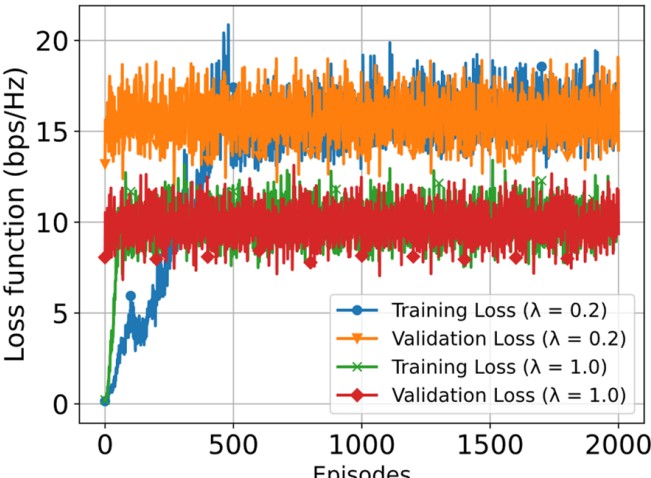

**Figure 11.** Loss function across the training and validation sets for two hyperparameter $\lambda$ values related to the total power constraint function. $P_s = 1.0$ W, $P_{av} = 3.0$ W. The relaxed $\lambda = 0.2$ allows the power constraint to be violated, resulting in higher loss values, while the stricter $\lambda = 1.0$ yields lower loss values. No overfitting is observed.

## 5. Conclusions

In this paper, several channel-model-free methodologies and heuristics are explored for optimal PA and then compared to the unparameterized solution for an all-FSO, multi-aperture satellite downlink between a GEO satellite and an OGS. A DRL-aided power allocation methodology is proposed for optical satellite systems disregarding any channel statistics knowledge requirements. Therefore, the PA problem is formulated as a constrained learning optimization problem with peak and total expected power inequality constraints. A power allocation algorithm was developed, aided by a DNN which is fed CSI observations and trained in a parameterized on-policy manner through a stochastic policy gradient approach. The proposed method does not require the channels' transition models or fading distributions. To validate and test the proposed allocation scheme, experimental measurements from the ARTEMIS optical satellite campaign were utilized. The proposed scheme performs 18% better than Equal Power, 16% better than DQN, and 53% better than Random Power, and it approximates the supervised method, with only 2% less accuracy, for $P_{av} = 2.0$ W. The predicted average channel powers match very well with the supervised ones, only differing by 0.1–0.2 W, while the allocated channel powers of the other schemes differ significantly more, about 0.1–0.5 W. Two hidden layers, 256 neurons, and two-parameter distributions are the optimal choices for the policy DNN. Finally, the proposed PA strategy is deemed appropriate for difficult optical conditions since it learns explicitly from observation.

**Author Contributions:** Conceptualization, T.T.K., N.K.L. and A.D.P.; methodology, T.T.K. and N.K.L.; software, T.T.K.; validation, T.T.K.; formal analysis, T.T.K.; investigation, T.T.K.; resources, N.K.L. and A.D.P.; writing—original draft preparation, T.T.K. and N.K.L.; writing—review and editing, A.D.P.; visualization, T.T.K.; supervision, A.D.P. All authors have read and agreed to the published version of the manuscript.

**Funding:** This research received no external funding.

**Institutional Review Board Statement:** Not applicable.

**Informed Consent Statement:** Not applicable.

**Data Availability Statement:** Data are contained within the article.

**Acknowledgments:** The work presented in this paper was carried out under the ONSET-CCN ESA project.

**Conflicts of Interest:** The authors declare no conflicts of interest.

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
