# Peer review of "Optimal Power Allocation in Optical GEO Satellite Downlinks Using Model-Free Deep Learning Algorithms"

_electronics, doi:10.3390/electronics13030647_

Round 1
Reviewer 1 Report
Comments and Suggestions for Authors
The article proposes a GEO satellite power allocation algorithm based on deep learning. The main idea is to determine the optimal power allocation for each channel at a certain capacity by perceiving the state information of the channel. This article has certain research significance, but the overall design needs to be supplemented with innovative and theoretical analysis and introduction. The main opinions are as follows:
1) In the GEO scenario, due to the relative immobility of the satellite relative to the Earth, the CSI under the satellite is relatively stable. Therefore, learning CSI should mainly focus on the initial moment in the early stage, and the subsequent use of deep learning has little performance gain. The author needs to explain in the text.
2) The author needs to further model the problem, which theoretically has not been solved by other optimization algorithms.
3) The author needs to add a beam description of the satellite and reflect it in the simulation. What is the bandwidth of each satellite? How is the coverage of satellites decided.
4) The simulation results lack mathematical statistics and some regular conclusions have not been drawn.
Comments on the Quality of English LanguageMinor editing of English language required
Author Response
Firstly, we would like to thank Reviewer 1 for his/her constructive feedback and we are grateful for acknowledging the concept of our contribution. In the revised version, all main opinions were considered and the necessary explanations have been inserted to make our contribution clearer and more comprehensive.
1) In the GEO scenario, due to the relative immobility of the satellite relative to the Earth, the CSI under the satellite is relatively stable. Therefore, learning CSI should mainly focus on the initial moment in the early stage, and the subsequent use of deep learning has little performance gain. The author needs to explain in the text.
Although in a LEO/MEO scenario, deep learning would be even more beneficial, the GEO-to-ground optical CSI is still unstable due to propagation effects that is the atmospheric turbulence effects that are induced in the slant path, i.e., the refractive index structure parameter varies due to the different temperature gradient, air pressure, and density along the altitude. Thin clouds may cause extra attenuation and even block the laser beam as well. Also, pointing and tracking errors cause time-varying fading in the received signal power. Therefore, it’s challenging to have a more accurate system model and fading distribution knowledge in the short and long term. Thus, the Authors believe that the proposed learning CSI algorithm has a great performance gain since it requires only CSI observations that can be measured.
2) The author needs to further model the problem, which theoretically has not been solved by other optimization algorithms.
Further modelling of the problem is out of the scope of the paper and shall be included in a future contribution since it requires further research and work. Although the optimization problem has been approached in the literature, it has never been presented in a MIMO optical satellite scenario along with available experimental data. The Authors have analytically stated the paper’s contributions (lines 96-120), which are sufficient for a Journal contribution.
3) The author needs to add a beam description of the satellite and reflect it in the simulation. What is the bandwidth of each satellite? How is the coverage of satellites decided.
Table 1 has been revised to provide an additional description of the ARTEMIS satellite laser beams. The bandwidth for a single optical beam (819 nm) is 2 Mbit/s in the forward direction (downlink) and 50 Mbit/s in the return direction (uplink). ARTEMIS has a communications coverage of Europe, Africa and the Middle East. In the simulations, only normalized irradiance data are employed to create a custom power allocation scenario. The reported wavelength, transmitted power, beam diameter, bandwidth, coverage are unnecessary for the simulation part. The transmitted powers are derived from the proposed algorithm, and we concentrate on the system capacity in bps/Hz.
4) The simulation results lack mathematical statistics and some regular conclusions have not been drawn.
When reinforcement learning techniques are proposed, the most considered performance metrics are the cumulative or discounted rewards, the convergence rate, the robustness, the benchmarks/baselines, the generalization on new unseen data, and the model complexity. These metrics have been used in our contribution. We can draw regular and significant conclusions by visually inspecting the discounted rewards over time and the allocated powers in Figures 4-10. The convergence rate can be estimated by the slope or asymptote of the learning curves. Moreover, the robustness and generalization on new data is expressed in Figure 11 while the time and spatial complexity are given in lines 343-345 and Table 3 accordingly. Finally, benchmarks/baseline algorithms are provided (Equal, Random, DQN, Supervised). There are many qualitative results that are presented and show the significance of the contribution.
We believe that the objective of the manuscript is clear and the revised version has been improved addressing all the comments of the Reviewer 1 and in its current form deserves publication in Electronics.
Reviewer 2 Report
Comments and Suggestions for Authors
This manuscript proposes a new method of DRL-aided algorithm to allocate power in an optical GEO-to-ground multi-aperture system optimally. The demonstration of the process is comprehensive. A detailed discussion of design and analysis is presented. The work is also validated using proper fabrication of the proposed design. In my opinion, the work is sufficient for publication. Therefore, there are several minor comments for this manuscript.
(1) In Figures 1, 6(a), and 10, the legend should be positioned to the right or below the graph, ensuring that the curve remains unobstructed as much as possible.
(2) Using vector graphics or high-resolution images is recommended for displaying coordinate diagrams, such as Figure 2, Figure 4, Figure 6, Figure 8, and Figure 11.
(3) Please double confirm the formulas, such as line 135 , and line 187. The symbol "â–¡" should be explained.
(4) I hope to see some qualitative observations of their results, except for Section 4 quantitative results.
(5) Please show the scalability of the designed model.
(6) It is suggested that authors should further emphasize the new or novelty of this proposed method.
(7) Please explain why the predicted average capacity greatly exceeds other baseline heuristic algorithms.
Comments on the Quality of English Language
Please improve the English writing.
Author Response
We would like to thank Reviewer 2 for his/her thoughtful comments on our manuscript and his/her reasonable inquiries. In the revised version, all the recommendations were considered.
(1) In Figures 1, 6(a), and 10, the legend should be positioned to the right or below the graph, ensuring that the curve remains unobstructed as much as possible.
Figures 1, 6(a), and 10 are now revised. The legends are now appropriately positioned to avoid obstruction of the curves.
(2) Using vector graphics or high-resolution images is recommended for displaying coordinate diagrams, such as Figure 2, Figure 4, Figure 6, Figure 8, and Figure 11.
Figures 2, 4, 5, 6, 7, 8, 9, 10, 11 are now revised. The image quality is now improved by using Scalable Vector Graphics (SVG).
(3) Please double confirm the formulas, such as line 135, and line 187. The symbol "â–¡" should be explained.
The mathematical formulas in lines 135 and 187 are now revised.
(4) I hope to see some qualitative observations of their results, except for Section 4 quantitative results.
Qualitative results: The proposed solution outperforms the three benchmark algorithms by showing a predicted average capacity advantage, and approximates the supervised method with excellent accuracy. The predicted average channel allocated powers match very well the supervised ones, i.e., more power is assigned to the channels with better conditions. The Trun. Normal distributions' parameters (mean and st. dev.) are stabilized according to channel conditions. Good propagation conditions indicate a higher mean and a smaller std. Also, no overfitting is observed and the hyperparameter λ can be selected to adjust the trade-off between the power constraint violation and loss function. Two hidden layers, 256 neurons, and two-parameter distributions are the optimal choices for the policy DNN. The parameter-sharing allows centralized training under a single policy for enhanced convergence and training speed.
Quantitative results: The proposed solution outperforms the Equal Power algorithm by 18%, the Random Power by 53%, the DQN by 16%, and exhibits good agreement with the supervised algorithm by 2%. The average channel allocated powers of the proposed algorithm vary approximately by 0.1-0.2 W the supervised ones, i.e., more power is assigned to the channels with better conditions. Meanwhile, the baselines vary a lot more by 0.1-0.5 W. The inference time of an episode is 0.7-0.8 s. The mean of ch.0 (~1.2) is greater than ch.3 (~0.85) and the st. dev. of ch.0 (~0.2) is smaller than ch.3 (~0.5). Thus, ch.0 has a much higher probability of being allocated with peak power than ch.3.
(5) Please show the scalability of the designed model.
The proposed learning model has been tested in scenarios with more than five optical satellite downlinks (higher-dimensional state-action space) and a greater amount of data and managed to retain its performance. Especially, the proposed DRL-aided solution is more scalable than its counterpart DQN algorithm because scalability is challenging for DQN due to the Q-table is not scalable when there are large, high-dimensional, and continuous state-action pairs.
(6) It is suggested that authors should further emphasize the new or novelty of this proposed method.
Although the optimization problem has been approached in the literature, it has never been presented in a MIMO optical satellite scenario along with available experimental data. The Authors have analytically stated the paper’s contributions (lines 96-120), which are sufficient for a Journal contribution.
(7) Please explain why the predicted average capacity greatly exceeds other baseline heuristic algorithms.
From lines 334-338: In Figures 4, 6, and 8 we observe that the proposed model-free DRL-aided solution outperforms the three baseline algorithms showing a predicted average capacity advantage (18% better than Equal Power, 16% better than DQN, 53% better than Random Power for ??? = 2.0 ?). Additionally, it approximates the supervised method with excellent accuracy (2% lesser performance for ??? = 2.0 ?).
We believe now that the objective of the manuscript is clear to the Readers and in its current form deserves publication in Electronics.
Reviewer 3 Report
Comments and Suggestions for Authors
In the manuscript, the author investigated using model-free deep learning algorithms to optimize the power allocation in the optical GEO satellite downlinks. The study is comprehensive, both including analytical and simulation data. Also, the results are well presented. Therefore, I do recommend this paper to be published with minor comments.
Some definitions in formula are not clear to me. For example, page 3 line 135 and page 5 line 187. Please verify.
Author Response
We would like to thank Reviewer 3 for his/her positive response for our manuscript and his/her helpful suggestions.
Some definitions in formula are not clear to me. For example, page 3 line 135 and page 5 line 187. Please verify.
The mathematical formulas in page 3 line 135 and page 5 line 187 are now revised.
We believe that the manuscript in its current form deserves publication in the Electronics Journal.